# *Solanum lycopersicum* Seedlings. Metabolic Responses Induced by the Alkamide Affinin

**DOI:** 10.3390/metabo11030143

**Published:** 2021-02-27

**Authors:** Tonatiu Campos-García, Jorge Molina-Torres

**Affiliations:** Department of Biotecnología y Bioquímica, CINVESTAV Unidad Irapuato, Irapuato, Guanajuato 36924, Mexico; tonatiu.campos@cinvestav.mx

**Keywords:** *Solanum lycopersicum*, alkamides, affinin, metabolite profiling, tomato, plant immunity, gas chromatography couplet to mass spectrometry (GC/EIMS)

## Abstract

Alkamides have been observed to interact in different ways in several superior organisms and have been used in traditional medicine in many countries e.g., to relieve pain. Previous studies showed that affinin when applied to other plant species induces prominent changes in the root architecture and induces transcriptional adjustments; however, little is known about the metabolic pathways recruited by plants in response to alkamides. Previous published work with Arabidopsis seedlings treated in vitro with affinin at 50 µM significantly reduced primary root length. In tomato seedlings, that concentration did not reduce root growth but increase the number and length of lateral roots. Non-targeted metabolomic analysis by Gas Chromatography couplet to Mass Spectrometry (GC/EIMS) showed that, in tomato seedlings, affinin increased the accumulation of several metabolites leading to an enrichment of several metabolic pathways. Affinin at 100 µM alters the accumulation of metabolites such as organic acids, amino acids, sugars, and fatty acids. Finally, our results showed a response possibly associated with nitrogen, GABA shunt and serine pathways, in addition to a possible alteration in the mitochondrial electron transport chain (ETC), interesting topics to understand the molecular and metabolic mechanisms in response to alkamide in plants.

## 1. Introduction

Plants respond to a wide variety of external cues which can be biotic or abiotic, and this response is mediated by molecular and metabolic rearrangements. Affinin (*N*-isobutyl-2*E*,6*Z*,8*E*-decatrienamide) is a low molecular weight α-unsaturated acyl-chain amide distributed in several plant species, however, affinin is the most abundant alkamide found in the ethanolic root extract from *Heliopsis longipes* (A. Gray) S. F. Blake, an endemic species from “Sierra Gorda”, Guanajuato, México. To date, its presence has not been detected in model plants such as Arabidopsis and tomato. It is known that this metabolite has a wide range of biological activities in bacteria, fungi, mammals and plants [1,2,3,4,5]. It has been found that alkamides, when applied to a non-alkamide-producing plant species, can alter plant signaling, modulating both developmental and stress response pathways [6,7]. Almost all organisms contain amide lipids composed of one or two amines linked to a fatty acid through an amide bond in their inner and outer membranes. For example, *N*-acylethanolamides (NAEs), widely distributed amide lipids, have important biological signaling functions in plants [8,9,10] and are important endocannabinoid signals in mammals [11,12]. *N*-acyl-homoserine lactones (AHLs) are an amino lipid compound that many bacteria use as quorum sensing signals to coordinate their collective behavior [13,14]. Additionally, some AHL prime plants for cell wall reinforcement and induce resistance to bacterial pathogens, via the salicylic acid/oxylipin pathway [15]. Affinin has shown a dramatic effect on *Arabidopsis thaliana* (Arabidopsis) root system architecture by altering primary root growth, increasing lateral root emergence and root hair elongation in a dose-dependent manner [3,7]. *N*-isobutyl decanamide (decanamide), a synthetic acyl amide obtained by the catalytic reduction of affinin, and the interacting signal nitric oxide (NO), act downstream independently of auxin-responsive gene expression to promote lateral root formation and emergence, providing evidence that NO is an intermediate in response to this acyl amide [16]. Furthermore, it was found that decanamide could modulate some necrotrophic-associated defense responses through jasmonic acid (JA)-dependent and MPK6-regulated signaling pathways [7]. In plants, fatty acid amide hydrolase (FAAH) overexpression, which is the amide hydrolase for *N*-acyl ethanolamides (NAEs), alters salicylic acid (SA) accumulation and signaling which in turn compromises innate immunity to bacterial pathogens, suggesting a functional mechanism of NAEs in plant immunity [17]. Alkamides like structures (NAEs and AHL) have been showed to exert a wide range of effects related to plant resistance due to changes in different metabolic pathways [10,15]; however, there is no information approaching the full metabolomic re-adjustment induced by natural unsaturated alkamides like affinin. The aim of this study was to give compelling evidence that unsaturated alkamides like affinin, induce a full metabolic adjustment associated to stress tolerance in a crop plant like *Solanum lycopersicum* L. and detect another metabolic pathway that could be a good prospect for further research. Results showed that in tomato seedlings, affinin at 100 µM significantly alter a wide range of metabolic pathways, mainly carbon, nitrogen, and energetic metabolism. Furthermore, plant response to affinin was tissue-specific and roots were the tissue with the most highly accumulated metabolites.

## 2. Results

### 2.1. Developmental and Metabolic Response of Solanum lycopersicum to the Alkamide Affinin

#### 2.1.1. Effect of Affinin in *Solanum lycopersicum* Development

To evaluate the effect of affinin in tomato seedlings, ten seedlings were grown under in vitro conditions for ten days in MS medium containing affinin at 0, 50 or 100 µM. Results showed that affinin treatments increase the number of emerged lateral roots (eLRs) and lateral roots length (LRL), although, affinin at 100 µM reduces primary root length (PRL). The higher affinin concentration (100 µM) reduce shoot fresh weight (FW), however, despite the reduction in primary root length, it does not have a significant effect changing root FW, while affinin at 50 µM did not had a significant effect on biomass accumulation (Figure 1).

#### 2.1.2. Metabolic Profiles Altered by Affinin

A non-targeted metabolomic profiling by GC/EIMS was used to appreciate the metabolomic response of *S. lycopersicum* to the alkamide treatment. A total of 55 non-redundant metabolites with known chemical structure were detected in shoots, and 60 in roots (Appendix A), of which 67% showed a match value of 900 or greater, 27% showed a match between 850–900 and 6% showed a match lower than 850 (Appendix A). Metabolites were then categorized into classes: sugars, sugar acids, phosphorylated compounds, organic acids, fatty acids, amino acids, and others, where amino acids were the main group of detected metabolites (Appendix A). Staked bars contain three colors from orange for R.Match >900, to yellow, for R.Match <850. The number of the total metabolites with this low R.Match value is small. Furthermore, a metabolic pathway analysis (MetPA) was made with MetaboAnalyst V4.0 using the differentially accumulated metabolites induced by affinin at 100 µM treatment and are presented in Figure 2, (a) shoots and (b) roots. In Figure 2a,b, the horizontal bar graph reveals the most significant pathways identified. Bars are colored based on their *p*-values where lower *p*-values are redder, and the bar length is based on the fold enrichment [18]. Results showed that affinin treatment significantly enrich different metabolic pathways in shoots and roots.

In shoots, the main enriched pathways include: “alanine, aspartate and glutamate metabolism”; “arginine biosynthesis”; “glyoxylate and dicarboxylate metabolism”; “carbon fixation in photosynthetic organism”; “glycine, serine and threonine metabolism”; as well as “starch and sucrose metabolism”; while enriched pathways in roots include: “aminoacyl-tRNA biosynthesis”; “glyoxilate and dicarboxylate metabolism”; “alanine, aspartate and glutamate metabolism”; “arginine biosynthesis”; and “glycine, serine and threonine metabolism”.

#### 2.1.3. Metabolic Variations Induced by Affinin Treatments

The presented results are differences based on relative abundance from the deconvoluted peak areas. Data from the peak areas of treatments and control plants were processed, normalized, and submitted to the statistical analysis. In metabolomics, it is often assumed, in metabolomic studies, that most of the observed changes in metabolite concentrations or spectral profiles are a result of normal physiological variations as background noise, and that only a small proportion of these changes are associated with the experimental condition of interest. Identifying these “key” features is typically the first step toward understanding the biological processes involved in the condition under investigation [18]. The statistical analysis performed allow us to identify key features that are involved in the condition under investigation. The present work did not use an internal standard; instead, it used the control plants as a background noise (normal metabolite variation) in order to detect variations in the metabolite relative abundance. For the identification of differentially accumulated metabolites with significant statistical differences, we used a univariate method called significance analysis of microarray approach (SAM) adapted for metabolomic analysis [18] in MetaboAnalyst V4.0 platform for shoots and roots (Table 1; Appendix A). Table 1 only shows metabolites that have statistical differences in their relative abundances based on SAM analysis, the ChEBI ID and the standard deviation (StDev) of the data. Results showed that, affinin at 100 µM induces a significant up-accumulation of several metabolites: thirteen metabolites in shoots and thirty-seven in roots (Table 1). Box and whisker plots of the normalized relative abundances from the metabolites detected with the SAM analysis are showed in Appendix A. We identified more metabolites that were significantly accumulated in roots (Table 1). Interestingly in both, shoots and roots, the relative abundance of fumarate is highly reduced while malate is increased (Table 1, Appendix A). To visualize differentially accumulated metabolites and correlations that could be present among factors, we used a multivariate exploratory data analysis. Data sets from shoots and roots were subjected to hierarchical clustering analysis and represented as a heatmap. This analysis revealed interesting patterns, in both shoots and roots. Most of the detected compounds were affected by affinin treatment in a dose-dependent manner (Figure 3). Metabolite accumulation induced by affinin in shoots represents the 62% of the total metabolites detected while in roots this is 80%. Amino acids are the main class of metabolites induced by affinin at 100 µM treatment, followed by organic acids, unsaturated fatty acids and other metabolites related to resistance against biotic and abiotic stress, such as myo-inositol, cadaverine, GABA, malate, pyroglutamic acid and chlorogenic acid.

#### 2.1.4. Global Metabolic Profile Changes Revealed by Partial Least Squares-Discriminant Analysis (PLS-DA) in Response to Affinin

To identify metabolites showing different relative abundance for each pairwise comparison, a partial least squares discriminant analysis (PLS-DA) was applied to highlight differences. The PLS-DA was performed based on the relative abundance of the deconvoluted chromatographic peaks of the identified metabolites on each tissue studied, shoot or root, for a model comparison between affinin-treated or control plants. Score scatter plots of the PLS-DA models show that the metabolite profiles of all plant samples were completely separated (Figure 4a,b), the first principal component (PC1) explains 41.3% and 53.2% of the variation in the shoot and root, respectively, while the second principal component (PC2) explains 12 and 12.3%, respectively. These results showed that affinin concentration has an impact on the metabolic response of tomato seedlings, which leads to the accumulation of several plant metabolites. In shoots, the most important compounds include fructose, asparagine, lactate, urea, phosphoric acid, linoleic acid, glucose, sucrose, stearic acid, glutamine and chlorogenic acid; while in roots these include malate, asparagine, glutamine, lactate, octopamine, melibiose, lysine, phosphoric acid, urea, valine and serine (Figure 4c,d). As might be expected, these compounds agree very well with the previous list generated by the univariate SAM model.

#### 2.1.5. Independent Metabolic Response from Shoots and Roots Induced by Affinin

To understand if there is a correlation between the two independent factors (tissue and affinin) on metabolite accumulation, we made a two-way ANOVA with the MetaboAnalyst V4.0 platform. The multivariate PCA analysis shows that the factor tissue had a major effect, grouping the accumulated metabolites, roots being the tissue that accumulated the higher level of metabolites.

Nevertheless, affinin treatments also were involved in separating the accumulated metabolites (Figure 5a). The Venn diagram shows that 56/66 (84.8%) of the metabolites displayed significant relationships with tissue, affinin treatment, or a tissue–treatment interaction (Figure 5b). Ultimately, tissue is the main factor that explains the variance in metabolite accumulation, with a lower metabolite content in the shoot represented by the larger blue area in Figure 5c.

Taking together all the above results, a core metabolism overview of metabolic changes in shoots and roots induced by affinin were constructed. Pathways were mapped with their corresponding metabolite and function in the Kyoto Encyclopedia of Genes and Genomes (KEGG) database for *Solanum lycopersicum* (Figure 6).

## 3. Discussion

### 3.1. Affinin Modulates Tomato Seedlings Development

The effect of affinin on plant development has been reported only for *Arabidopsis thaliana* seedlings and at the transcriptional level. In Arabidopsis, affinin treatments greater than 50 µM have a significant inhibitory effect on primary root growth [3,6], interestingly, those concentrations of affinin do not reduce the PRL in tomato seedlings (Figure 1a). However, higher affinin concentrations increase the number of eLRs in tomato seedlings. The effect of affinin on LRL has not been detailed, nevertheless, decanamide treatments greater than 56 µM have been shown to have a significative inhibitory effect on LRL from Arabidopsis seedlings [6]; in contrast, our results showed that for tomato seedlings, affinin at 50 and 100 µM increases LRL (Figure 1a). These differences in tomato tolerance to the inhibitory effect of high affinin concentration could be related to the physical structure of tomato roots. It is known that domesticated tomato cultivars have an increased number of roots cells and cortex layers, in addition to the fact that in flowering plant species, the outer layer of the root cortex, or exodermis, contains a suberized cell wall to restrict the passage of solutes from the outside of the root to the inside, but Arabidopsis does not present that suberized exodermis [19]. Nevertheless, despite affinin at 100 µM reduced shoot FW, root FW was not significantly reduced (Figure 1b). This effect could be explained by the fact that affinin increases the eLRs and LRL; therefore, lateral roots compensate root biomass despite the shorter root length. Our results demonstrate that the effects of affinin on tomato are like the effects induced on Arabidopsis in terms of root development. However, tomato shows to be more tolerant to the inhibitory effect of high concentrations of affinin.

### 3.2. Affinin Induces Metabolic Adjustment in Tomato Seedlings

In plants, metabolic adjustment is vital for adaptation to environmental biotic or abiotic stress. The maintenance of metabolic balance and the accumulation of certain metabolites have been associated with tolerance to diverse stresses in different plant species [20,21,22,23]. Our results showed that affinin treatment at 100 µM triggers the accumulation of metabolites in both shoot and roots. In Table 1 are shown only the metabolites that have statistical differences in their relative abundance based on SAM analysis, the ChEBI ID and the Standard Deviation (StDev) of the data analyzed with the SAM algorithm. Many of those metabolites have been reported to be involved in different metabolic pathways related to plant adaptation and resistance to different plant stresses. The biological functions of metabolites differentially accumulated by affinin that could be involved in tolerance to different stresses are discussed below.

#### 3.2.1. Affinin Alters Sugar Metabolism

Several studies have provided hints supporting a function of primary metabolism in regulating known defense pathways in plants [24]. In shoots, affinin induces the up-accumulation of sugars like fructose, glucose and sucrose, while melibiose was down accumulated. This result suggests that, in shoots, affinin could be activating melibiose degradation to produce glucose. Melibiose degradation is a single-step pathway, where melibiose is degraded via α-galactosidase to galactose and glucose [25]. Our analysis shows that in roots, melibiose and fructose were up accumulated. The presence of high amounts of melibiose in roots has been associated with enhanced root colonization by rhizobacteria [26]. Together, these results suggest that affinin alter primary metabolism, and the up-accumulation of sugars like glucose in shoots could be associated to plant resistance against pathogens, and the increase in melibiose in roots could stimulate the beneficial bacteria association.

#### 3.2.2. Affinin Alters Amino Acid Metabolism

Amino acids are the building blocks of proteins, nevertheless, the role played by accumulated amino acids in plants varies from acting like osmolytes to the regulation of ion transport, modulating stomatal opening, the detoxification of heavy metals, synthesis and activity of some enzymes, gene expression and redox homeostasis [27,28]. Our results, summarized as the core metabolic changes by affinin treatment in Figure 6, show that several amino acids, in shoots and roots, are up accumulated by affinin at 100 µM treatment. However, the cluster analysis and the heatmap show that in shoots, at least 12 amino acids were up accumulated. The statistical analysis points to that only three of them (Asn, Glu, Gln) had significative differences. In roots, we found that 17 amino acids were significantly up accumulated. It has been found that during interaction with pathogens, the host glutamate metabolism is markedly altered, leading to a metabolic state, termed “endurance”, in which cell viability is maintained, and this modulation results in resistance to necrotrophic pathogens [29]. On this matter, affinin at 100 µM induces a significant accumulation of Glu, Gln and several amino acids related to nitrogen metabolism. It has been showed that the affinin-related molecule, decanamide, increases NO accumulation in roots [16]. Together, these results suggest that alkamides could be activating nitrate reductase (NR) that use NO2^−^ to produce NO. The activation of nitrogen metabolism, is reflected in higher levels of major amino acids including GABA, Ser, Gly, Glu, Gln, etc. The accumulation of Ala induced by affinin in both tissues suggests that Ala could act as a nitrogen reservoir to feed the TCA cycle as well as amino acids’ biosynthesis [30,31]. In tomato roots, affinin at 100 µM treatment increases the abundance of key metabolites in Ser biosynthetic pathways, phosphorylate and glycerate pathways, like glycerate, 3-phosphoglycerate, Gly and Ser [32,33]. In contrast, in tomato shoots, there is a reduction in Ser while glycerate is up accumulated. In roots, phosphorylate and glycerate pathways represent the only ways of Ser biosynthesis. However, these two pathways have not been extensively studied in plants until recently. It is interesting that affinin treatment may have an impact on them. In addition, we found that ethanolamine is up accumulated by affinin treatment in both shoot and roots. This is noteworthy because it is well known that ethanolamine is an NAE derivative [9], which is an important lipid-derived cell signaling mediator in plants and mammals. In roots, shikimate was up accumulated by affinin at 100 µM, followed by an increase in the aromatic amino acid Phe while Tyr was down accumulated in shoots and was not found in roots (Appendix A). Phe and Tyr are synthetized from chorismite, the end-product of the shikimate pathway. In plants, the phenylpyruvate pathway can utilize a cytosolic aminotransferase that links the coordinated catabolism of Tyr to serve as the amino donor to produce Phe [34]. Taking account these findings, it can be suggested that affinin could be altering the phenylpropanoid metabolism leading to the catabolism of Tyr and increasing the abundance of Phe and chlorogenic acid (Figure 6, Appendix A). All the amino acids involved in the Asp-family pathways (Lys, Thr, Met and Ile) are up accumulated by affinin at 100 µM; suggesting that this treatment has an important impact in the energy metabolism-associated network [35]. In addition, we found that cadaverine, a Lys derivative, has increased content in tomato roots after affinin at 100 µM treatment and is only found in roots, which agrees with what was previously reported, that cadaverine was only found in roots and its presence has been related to plant resistance to stress conditions [36,37]. The branched chain amino acids (BCAAs) Val and Leu serve as alternative energy sources [38,39]. In response to affinin, shoots tend to accumulate Val but Leu is maintained at control levels. This is interesting because Leu catabolism provides an alternate source of acetyl–CoA to sustain respiration and metabolic processes in the absence of photosynthesis [38]. In contrast, in non-photosynthetic tissues, as in roots, we found that affinin significantly increases the accumulation of both BCAAs, Val and Leu.

#### 3.2.3. Organic Acids and Sugar Acids Accumulated by Affinin Treatments

Experimental evidence has associated organic acid metabolism with plant tolerance to abiotic stresses, like nutrient deficiencies, metal tolerance and plant–microbe interactions operating at the root–soil interphase [40].

Our results showed that, in roots, affinin treatment induces a significative up-accumulation of organic acids and sugar acids like glycerate, gluconate, lactate, malate, succinate and isocitrate (Figure 6, Appendix A). We found that affinin at 100 µM significantly increases gluconate content, while it reduces the glucose content, suggesting that the gluconate shunt [41,42] could be activated by affinin. Furthermore, an increase in gluconate in plants has been associated to the effect of humic acids, beneficial bacteria inoculation, and tolerance to drought and salinity [43,44,45]. Additionally, affinin treatment induces the accumulation of glycerate, 3-phosphoglycerate (glycerate 3P) and Ser, suggesting that the glycerate-serine pathway could be activated by affinin treatment. The sugar acid threonate was up accumulated in roots suggesting an induction of ascorbate catabolism leading to an increased level of threonate. Threonate is highly related to ascorbate metabolism, and it has been demonstrated that in tomato leaves, ascorbate degradation leads to threonate accumulation [46].

Organic acids such as isocitrate, succinate and malate are key metabolites in the core of TCA cycle and carbon metabolism. We found that tomato seedlings treated with affinin at 100 µM, significantly accumulate succinate in roots. Fumarate is significantly reduced and in both shoots and roots. Quantities of glycolate and malate are also up accumulated, suggesting the activation of an anaplerotic pathway supplying malate. These results are interesting by the fact that in roots, malate has been associated with different physiological responses as tolerance to heavy metals and increasing nitrogen fixation increase *Rhizobium* symbiosis, while in leaves maintain pH homoeostasis [40]. Succinate is involved in the electron transport chain (ETC) and in the ETC complex, being oxidized to fumarate by the enzyme succinate dehydrogenase (SDH) [47]. A dysfunction in SDH causes a disorder in the mitochondrial metabolism via the accumulation of succinate leading to a decrease in fumarate levels [47,48]. It is suggested that NO inhibits SDH at its UQ site or at its Fe–S centers [49]. Taking account these results, we propose that affinin could be inducing changes in the TCA pathway and NO accumulation, altering the SDH and mitochondrial ETC complex II activities, leading to an alternative route like a glyoxylate cycle to produce malate and the GABA shunt to synthesize succinate.

Succinate formed in the GABA shunt could be converted to isocitrate by the cytosolic isocitrate lyase, which, in turn, is converted to 2-oxoglutarate by isocitrate dehydrogenase [33] contributing to the redox homeostasis. All together, these results show that, in roots, many metabolites from the TCA cycle were up accumulated by affinin treatment and these changes coincide with a decrease in sugars as sucrose and glucose. In contrast, in shoots, a decrease in metabolites like Ser, Gly, glycolate, succinate and fumarate, suggest that, in photosynthetic tissues, the catabolism of these metabolites could be increased or these metabolites are being transported to sink tissues. In addition, affinin induces an increased rate of glycolysis which provides the carbohydrates for the TCA cycle, supplying the precursors for other reactions such as amino acid and organic acid synthesis as well as chemical energy like ATP, NADPH and NADH. In addition, results also suggest an activation of different alternate metabolic pathways, like GABA shunt and phosphorylate/glycerate-serine pathways.

#### 3.2.4. Fatty Acids Accumulated in Response to Affinin

Fatty acids are an important source of energy reserve and vital components of membrane lipids in all living organisms. In Arabidopsis, it was found that decanamide induces plant resistance against *Botrytis cinerea* through the JA-dependent pathway and MPK6 signaling pathways [7]. Our results show that affinin treatment increases the accumulation of stearic acid (C18:0), oleic acid (C18:1) and linoleic acid (C18:2). These fatty acids are primarily found in plasma, thylakoids, and mitochondria membranes [50] and have been associated with resistance against *Diaporthe phaseolorum* (C18:0) [51], SA-mediated plant defense signaling (C18:1) [52] and basal defense against fungal pathogens like *Botrytis cinerea* (C18:2) [53]. In addition to its origin from the cell membranes, these fatty acids could be synthetized by the conventional Δ6-pathway [54], where stearic acid is the precursor of oleic acid which is further converted to linoleic acid and leads to the formation of linolenic acid, the precursor of JA. In summary, these results are consistent with the findings of Méndez-Bravo et al. (2011) [7]: that a lipid signaling pathway is being activated in response to alkamides, and that confers plant resistance against biotic stress.

#### 3.2.5. Differences in Roots and Shoots Metabolic Profiles in Response to Affinin Treatment

It is well known that in plants, roots take up nutrients and water from soil, while shoots metabolize these nutrients and produce different metabolites, and part of these metabolites are re-translocated to the roots. Many findings indicate that the biosynthetic and bioactive capabilities of roots are as diverse and complex as those of any other part of the plant [55]. Our results show that there is a tissue-specific up-accumulation of metabolites; however, root is the tissue with most metabolites up accumulated. Affinin treatment also has an effect on the type of tissue-specific metabolites accumulated as shown in the two-way heatmap visualization of metabolite abundance in roots, purple box, as well as in shoots, green box. Fumarate was significantly down accumulated in both tissues in response to affinin, yellow box (Figure 5c). The down-accumulation of this metabolite in affinin treated seedling and the up-accumulation of succinate, suggest that affinin could be inducing the alteration of the of SDH activity in the ETC complex II, reducing the abundance of fumarate and activating an anaplerotic pathway to supply malate into the TCA cycle, plus the activation of the GABA-shunt increasing the biosynthesis of succinate. Another new finding is that Gln is specifically accumulated in shoots while Glu is accumulated in roots and both are up accumulated due to affinin treatment. Attention-grabbing because it has been demonstrated that in Arabidopsis, Glu inhibits primary root growth and stimulates the outgrowth of lateral roots [56].

## 4. Materials and Methods

### 4.1. Plant Material and Growth Conditions

Two hundred tomato seeds of *Solanum lycopersicum* L. cv. Río Grande (McKezie Seeds, Brandon, MB, Canada; https://mckenzieseeds.com/, 15 January 2019) were germinated in square Petri dishes. Seeds were first sanitized by immersing for 5 min in a 20% sodium hypochlorite solution, then rinsed three times with sterile distilled water, and immersed for 5 min in a solution composed of Triton X-100 (2%) in 70% ethanol followed by ethanol 96% for 1 min and rinsed with sterile water. The germination medium was a solid salts MS [57] (0.3×) supplemented with sucrose (11 g L^−1^) and Agar (9 g L^−^^1^). Seeds were sown and left 72 h in the dark at room temperature. After this time, the seeds were left to germinate on vertically placed Petri dishes under light for 4 days at room temperature. When germination started, plants with a similar root length (1 cm) were selected and transferred to treatments. For the treatments, the same solid growth medium was prepared with 0.5× MS salts and the affinin treatments (0, 50 and 100 µM) were incorporated. During the experimental conditions, seedlings were grown under a long-day photoperiod (16 h light, 8 h darkness), 25 °C/18 °C day/night temperature and light intensity of 100–200 µM m^−2^ s^−1^. For experiments, 10 seeds were sown per treatment, with 3 treatments with 3 replications. Seedlings were left in treatments for ten days. For plant growth analysis, experiments were performed in duplicate with similar results and representative results are presented.

### 4.2. Plant Growth Analysis

Measurements of primary root length were performed on images, taken from the plates ten days after treatments (d.a.t.) using ImageJ software (www.imageJ.net, 15 January 2019). For the analysis of emerged lateral roots, ten days old root samples were visualized with a Zeiss stereo microscope model 2000-C equipped with a model ERc 5s Zeiss Axiocam digital camera (www.zeiss.com/, 23 March 2019). Roots protruding beyond the epidermal tissue were scored as emerged lateral root. For each treatment, at least 10 seedlings were analyzed. Roots and shoot (hypocotyl + cotyledons) were excised with a shaving blade and weighed, then frozen in liquid nitrogen and stored at cold temperature until they were used for further analysis. Experiments were performed in duplicate with similar results and the results from the second experiment were chosen as representative.

### 4.3. Non-Targeted Metabolic Profiling by Gas Chromatography—Mass Spectrometry (GC/EIMS)

Seedlings from ten days after treatments were frozen before analysis. Frozen samples were weighed (300 mg of shoot and 200 mg of roots), then grounded in a mortar with pistil adding a 80% (*v*/*v*) methanol:water. The extraction mixture was collected in a 15 mL Falcon tube, vortexed 1 min and then sonicated in a water bath at 40 °C for 30 min. Finally, the samples were centrifugated at 14,000 rpm for 5 min, the supernatant was poured in fresh Falcon tubes and stored at 4 °C. A 2 mL aliquot of each extract was taken to dryness in an Eppendorf Vacufuge^®^ plus device (45 °C) (www.eppendorf.com, 13 December 2020) and then resuspended in 200 µL of absolute methanol. Samples where dried and derivatized with 20 µL of pyridine plus 80 µL of BSTFA and incubated 30 min at 80 °C in an Eppendorf ThermoMixer^®^ (Eppendorf AG, Hamburg, Germany). After incubation, 100 µL of isooctane were added to each sample and transferred to chromatography vials with inserts. Samples were analyzed with an Agilent GC model 7890A coupled to an Agilent model 5975 with electronic impact ionization unit and quadrupole mass spectrometer and triple-axis detector, an Agilent model 7693A autosampler and an injector, equipped with a capillary column DB1MSUI (60 m, 250 µm, 0.25 µm. J&W, Agilent Technologies, Inc. USA). A 1 µL aliquot of the trimethylsilylated (TMS) samples were injected in pulsed split mode. Injection temperature was 230 °C. Helium was used as carrier gas with a constant flow of 1 mL min^−1^. The GC oven program started at 70 °C and held for 5 min, then increased at 5 °C min^−1^ to 280 °C and held for 15 min. The transfer line temperature was 250 °C. Temperature of the ion source and the quadrupole was 230 and 150 °C, respectively. The GC/EIMS method conditions were modified from Weckwerth et al. [58]. Measurements were performed in SCAN mode with a mass range from 50 to 800 *m/z* and mass spectra were obtained at 70 eV. Data were collected with the Mass Hunter Workstation version B.06.00 software (J&W, Agilent Technologies, Inc., Wood Dale, IL, USA). Retention time, purity of the peak and mass spectrum of each component were determined with the Automated Mass Spectral Deconvolution and Identification System “AMDIS” software version 2.66 (http://www.amdis.net/, 13 December 2020) and each compound was identified using the mass spectra database and library of the National Institute of Standards and Technology (NIST) MS Search software version 2 (Gaithersburg, MD, USA) and/or comparing mass spectra with the Golm Metabolome Database (http://gmd.mpimp-golm.mpg.de/, 18 November 2020). The peak areas obtained in AMDIS were normalized by weight. For non-targeted metabolomics, only one experiment with three biological replicates were made.

### 4.4. Data Processing and Statistical Analysis

For seedling growth measurements, data were analyzed with one-way ANOVA (*n* = 10) and a Tukey test (*p* ≤ 0.05) using the InfoStat software version 2017.1.2 (www.infostat.com.ar, 16 September, 2020). To investigate the differentially accumulated metabolites, the relative peak area was obtained, and normalized by dividing the compound peak area by the fresh weight of the sample. Data were transformed with the cube root transformation and scaled by Pareto algorithms, then data were subjected to a partial least squares–discriminant analysis (PLS-DA) using MetaboAnalyst software version 4.0 (www.metaboanalyst.ca/, 21 June 2020) a comprehensive tool suite for metabolomic data analysis [18,59]. The heatmap was also generated following this tool using Pearson as a cluster distance measure, Ward clustering algorithm and scaled by compound/feature. The metabolomic pathway analysis was constructed only using the highly accumulated metabolites with affinin at 100 µM in MetaboAnalyst 4.0 and the MetPA web-based tool dedicated to the analysis and visualization of metabolomic data within the biological context of metabolic pathways [60]. Pathway analysis algorithms used were the Fisher exact test and relative betweenness centrality. Bar plots were built with the ggplot2 package with R software version 4.0.2 (http://www.r-project.org/, 21 June 2020) and R studio version 1.3.959 (R Studio, Boston, MA, USA). The VIP scores graph that shows the most important or informative compounds were selected using the PLS-DA three-component model in MetaboAnalyst 4.0. For the identification of different statistically accumulated metabolites we used the significance analysis of microarray (SAM) approach, originally designed for microarray data analysis but that can also be used for metabolomic analysis [18]. SAM was designed to address the issue that in high-dimensional data analysis, the estimate of the variance tends to be unstable when the sample size is small, from 3 to ~8 per group [18]. For the two-way ANOVA analysis to compare shoot and roots metabolites, data sets were transformed with the generalized logarithm transformation algorithm and scaled by the Pareto scaling algorithm in MetaboAnalyst 4.0. Results are represented by 3D-PCA, the Venn diagram and heatmap with Pearson distance and average clustering algorithm. ANOVA type III with *p*-value ≤ 0.05 and multiple testing correction with a False Discovery Rate (FDR) were used as parameters for the two-way ANOVA.

## 5. Conclusions

We demonstrated that treated tomato seedlings are more tolerant to the dose dependent growth inhibition induced by affinin in Arabidopsis. We found a differential alteration of metabolites induced by affinin treatments in a dose dependent manner, in both, shoots and roots. The metabolome from shoots and roots shows an interesting pattern in response to affinin treatments, where amino acids, organic acids, sugars, sugar acids, and fatty acids are the major groups of metabolites altered. The metabolic compositions of roots and shoots were similar, nevertheless, the two-way ANOVA showed that roots accumulate more metabolites than shoots. Thus, there is clear evidence in the differences in resource allocation in both tissues by affinin treatment. However, in roots, more metabolites were significantly altered than in shoots. Finally, we found a response possibly associated with nitrogen, GABA shunt and serine pathways, in addition to a possible alteration in the mitochondrial electron transport chain (ETC), an interesting topic for further research to fully understand the molecular and metabolic mechanisms in alkamide sensing in plants.

## Figures and Tables

**Figure 1 metabolites-11-00143-f001:**
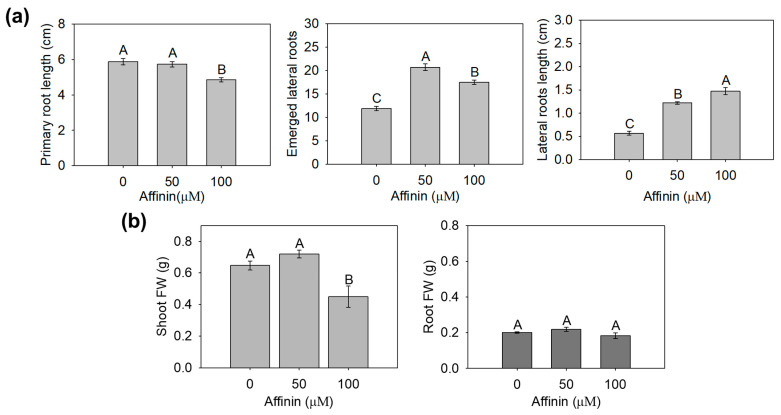
Changes in tomato seedlings root growth and morphology, induced by affinin treatment. Tomato seedlings were grown in a medium with or without the indicated affinin concentrations. After ten days, seedlings showed changes in: (**a**) primary root length (PRL), number of emerged lateral roots (eLRs), lateral roots length (LRL); (**b**) biomass expressed as fresh weight (FW) of five pooled shoots or roots with three biological repeats. Data were analyzed with one-way ANOVA and a Tukey test. Different letters (A–C) were used to indicate means that differ significantly (*p* ≤ 0.05) (*n* = 3).

**Figure 2 metabolites-11-00143-f002:**
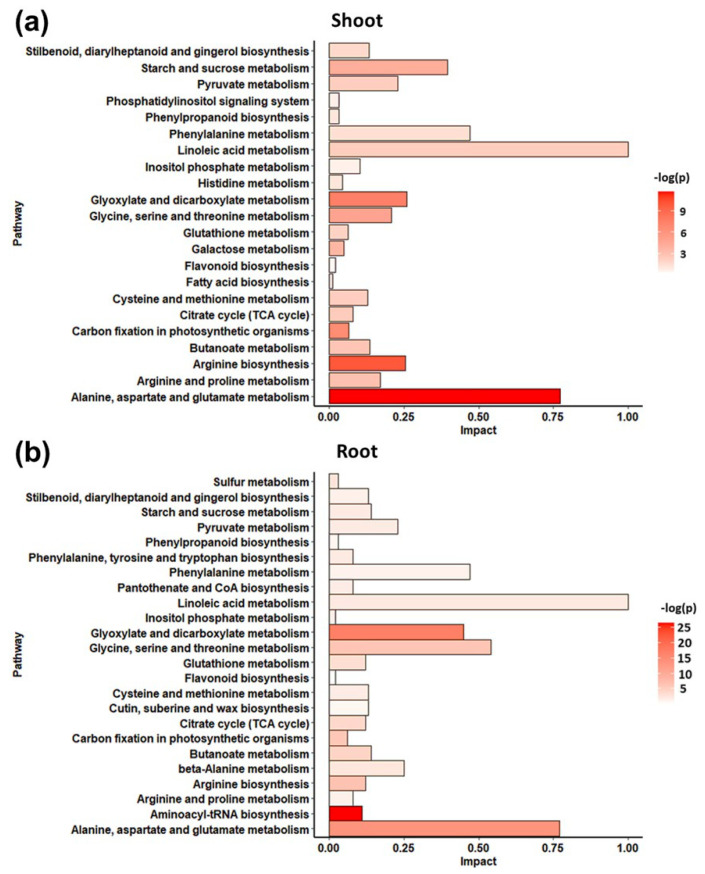
Enrichment pathway analysis. Metabolite pathways enrichment analysis of highly accumulated metabolites in (**a**) shoots and (**b**) roots from plants treated with affinin at 100 µM (*n =* 3).

**Figure 3 metabolites-11-00143-f003:**
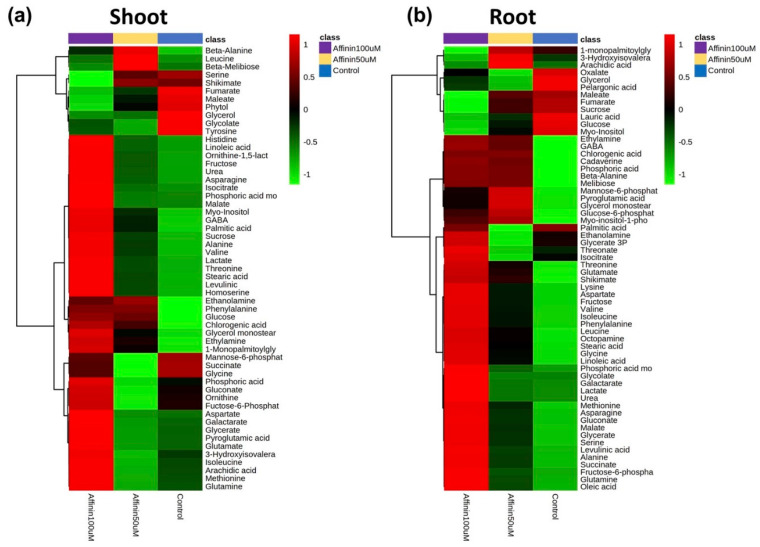
Heatmaps of changes in the average metabolite abundance in tomato (**a**) shoots and (**b**) roots in response to affinin treatments. Mean metabolite abundance for each sample type is shown: red, higher abundance; green lower abundance (*n =* 3).

**Figure 4 metabolites-11-00143-f004:**
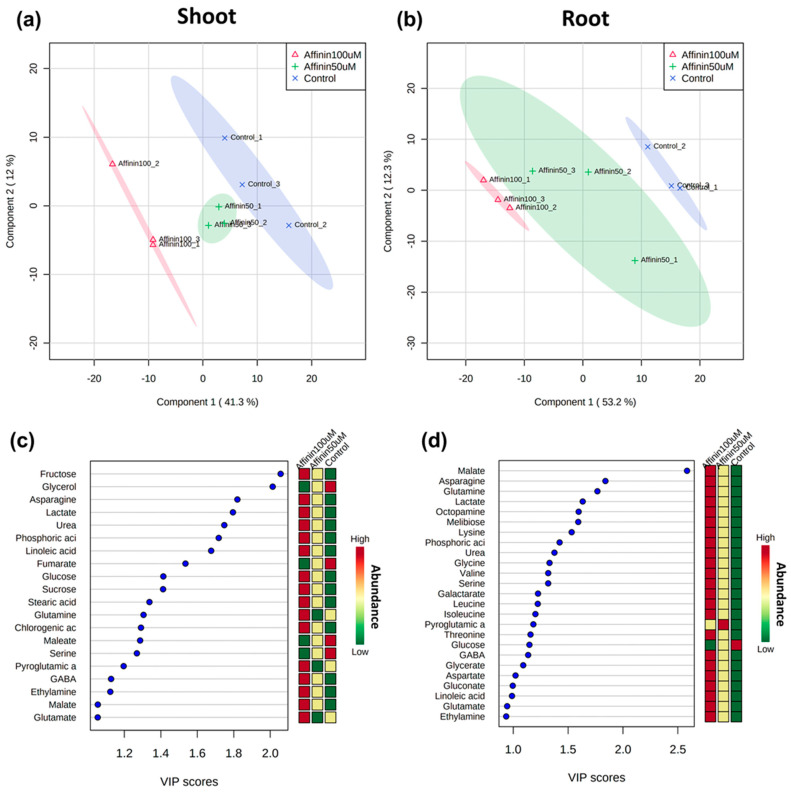
Score scatter plots of the two-component PLS-DA model based on the relative level of identified metabolites among samples from: (**a**) shoots R^2^ = 0.99, Q^2^ = 0.63; and roots (**b**) R^2^ = 0.96, Q^2^ = 0.79. The discrimination between affinin treatments for most important metabolites (VIP scores) responsible for the separation of clusters in PLS-DA, with the mini-heatmap on the right of each graph indicating their variation in concentration within different treatments for: shoots (**c**) and roots (**d**) (*n =* 3).

**Figure 5 metabolites-11-00143-f005:**
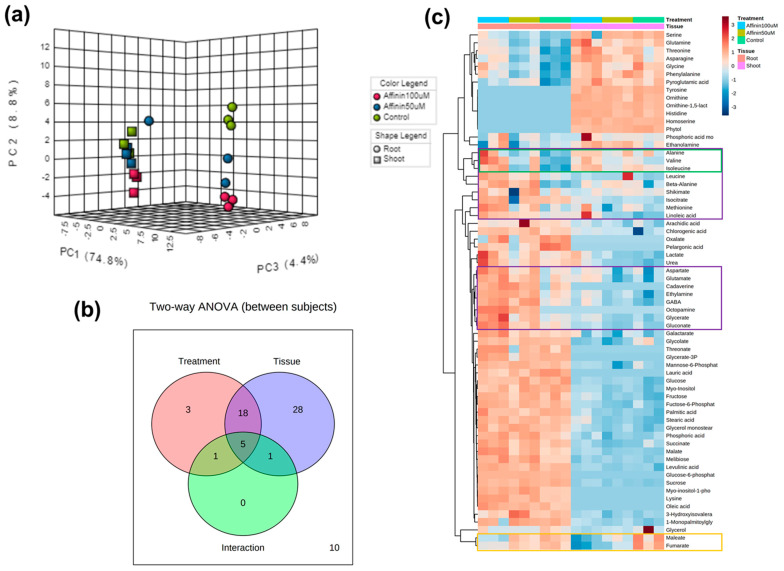
Two-way ANOVA visualization to identify interesting patterns between shoots and roots: (**a**) 3D scores scatter plot of PCA for the top three principal components, and the two experimental factors are indicated using different colors for treatments and different shapes for tissue (*n =* 3); (**b**) Venn diagram that summarizes the number of significant metabolites associated with each tissue, as well as their interactions; (**c**) two-way heatmap visualization of metabolite abundance. Distance was measured using the Pearson algorithm and the clustering algorithm using average distance.

**Figure 6 metabolites-11-00143-f006:**
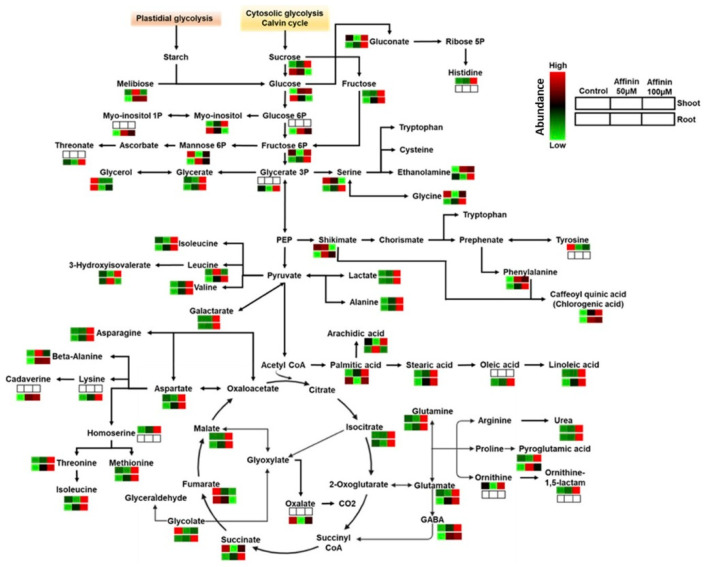
Core metabolism overview of metabolic changes in shoots and roots induced by affinin treatments. Pathways were constructed mapping each metabolite and function in the Kyoto Encyclopedia of Genes and Genomes (KEGG) database for *Solanum lycopersicum*-annotated pathways. Mean metabolite abundance for each sample type is showed in red for higher abundance or green for lower abundance while empty boxes indicate metabolites not detected.

**Table 1 metabolites-11-00143-t001:** Metabolites that show significantly higher abundances in both, shoot and roots from tomato seedlings treated with affinin, using the statistical significance analysis of microarray (SAM) approach.

Metabolite	ChEBI ID	StDev
SHOOTS		
Asparagine	17,196	6.4627
Chlorogenic acid	16,112	4.0187
Fructose	28,757	2.9712
Glucose	17,634	3.9011
Glutamate	16,015	1.61
Glutamine	18,050	13.8
Lactate	24,996	6.1811
Linoleic acid	17,351	5.1357
Phosphoric acid monomethyl ester	340,824	23.675
Pyroglutamic acid	16,010	8.5086
Stearic acid	28,842	1.8317
Sucrose	17,992	3.7142
Urea	16,199	3.2501
ROOTS		
Alanine	15,570	1.2658
Asparagine	17,196	3.1375
Aspartate	22,660	1.0977
Beta-alanine	16,958	0.19073
Cadaverine	18,127	1.3855
Chlorogenic acid	16,112	3.8009
Ethylamine	15,862	0.84901
GABA	16,865	1.2353
Galactarate	30,852	1.375
Gluconate	86,359	0.41726
Glucose-6-phosphate	14,314	1.3352
Glutamate	16,015	1.5105
Glutamine	18,050	3.1626
Glycerate	16,659	1.2093
Glycerate 3P	17,050	1.8376
Glycine	15,428	1.6957
Isocitrate	30,887	6.5588
Isoleucine	27,730	1.6544
Lactate	24,996	13.919
Leucine	25,017	1.114
Linoleic acid	17,351	0.59333
Lysine	18,019	1.2269
Malate	6650	4.1396
Melibiose	28,053	7.8211
Methionine	16,811	0.67585
Myo-inositol-1-phosphate	18,297	0.89362
Octopamine	17,134	0.12966
Oleic acid	16,196	2.1569
Phenylalanine	17,295	0.8976
Phosphoric acid	26,078	5.5247
Pyroglutamic acid	16,010	1.5695
Serine	17,822	3.5762
Succinate	15,741	1.5881
Threonate	15,908	2.5509
Threonine	16,398	0.51273
Urea	16,199	9.7979
Valine	27,266	1.0954

## Data Availability

The data presented in this study are openly available in FigShare at https://doi.org/10.6084/m9.figshare.14115575.v1.

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
