# Peer review of "Solanum lycopersicum* Seedlings. Metabolic Responses Induced by the Alkamide Affinin"

_metabolites, 2021, doi:10.3390/metabo11030143_

Round 1

Reviewer 1 Report

See File attached

Author Response

Thank you for your very interesting and substantial comments

Reviewer 2 Report

The manuscript entitled “Solanum lycopersicum metabolic profiling induced by the alkamide affinin reveal a response mechanism associated to carbon, nitrogen and energetic metabolism” represents a comprehensive metabolome study. Metabolites induced in many metabolic pathways were determined by a simple GC-MS procedure using derivatization with BSTFA. Next many relevant ideas were suggested on the basis of statistical analysis of roots and shoots samples chemical profiling data. To provide good stability of chromatographic measurements needed for further data analysis, one should use a validated method with good precision and accuracy. No reference showing necessary information about the GC-MS method used is given. Moreover, the use of methoximation prior to silylation is extremely important when working with polar organic acids, while there is no methoximation stage described in the part 4.3. There are many scientists who have great concerns about the non-targeted metabolomics studies due to the problems of uncertainties caused by such insufficiently elaborated methods and by metabolite identification using various databases (without satndards). However, it can be seen that the authors have identified a set of well-known metabolites, and the extend of certainty can was increased by using additional metabolome databases. Considering what was mentioned above about derivatization and GC-MS, one may disregard such potential biases due to the fact that many biological replicates were analyzed. Thus, the results and conclusions of the study can be considered as fully supported by the submitted data.

Some technical remarks:

  • Plant species should be given in italics
  • mL instead of ml
  • Méndez-Bravo et al. (2011) 347 findings [7]
  • Trimethylsilylated instead of Timethylsilylated

Author Response

We thank you reviewing the manuscript and your comments.  Best regards.

Round 2

Reviewer 1 Report

The manuscript was appropriately revised by the authors.

Concerning the title, my suggestion was unclear. The authors could keep a title such as "The alkamide affinin induces a broad metabolic response in Solanum lycopersicum seedlings".

Reviewer 2 Report

The authors have revised the manuscript to address all my questions and concerns. I recommend to accept it for publication in Metabolites.